# Long-Term Medical Resource Consumption between Surgical Clipping and Endovascular Coiling for Aneurysmal Subarachnoid Hemorrhage: A Propensity Score–Matched, Nationwide, Population-Based Cohort Study

**DOI:** 10.3390/ijerph18115989

**Published:** 2021-06-02

**Authors:** Yang-Lan Lo, Zen Lang Bih, Ying-Hui Yu, Ming-Chang Li, Ho-Min Chen, Szu-Yuan Wu

**Affiliations:** 1Department of Neurosurgery, Lo-Hsu Medical Foundation, Lotung Poh-Ai Hospital, Yilan 256, Taiwan; lyl3481@gmail.com; 2Department of Emergency Medicine, Lo-Hsu Medical Foundation, Lotung Poh-Ai Hospital, Yilan 256, Taiwan; turrando@mail.pohai.org.tw; 3Department of Colorectal Surgery, Lo-Hsu Medical Foundation, Lotung Poh-Ai Hospital, Yilan 256, Taiwan; yinghui0704@yahoo.com.tw (Y.-H.Y.); fairytale1208@hotmail.com (M.-C.L.); 4Department of Food Nutrition and Health Biotechnology, College of Medical and Health Science, Asia University, Taichung 413, Taiwan; homin.chen@gmail.com; 5Big Data Center, Lo-Hsu Medical Foundation, Lotung Poh-Ai Hospital, Yilan 256, Taiwan; 6Division of Radiation Oncology, Lo-Hsu Medical Foundation, Lotung Poh-Ai Hospital, Yilan 256, Taiwan; 7Department of Healthcare Administration, College of Medical and Health Science, Asia University, Taichung 413, Taiwan; 8Graduate Institute of Business Administration, Fu Jen Catholic University, Taipei 242062, Taiwan; 9Centers for Regional Anesthesia and Pain Medicine, Wan Fang Hospital, Taipei Medical University, Taipei 110, Taiwan

**Keywords:** medical reimbursement, hospital stay, ICU stay, surgical clipping, endovascular coiling, aneurysmal SAH

## Abstract

Purpose: To estimate long-term medical resource consumption in patients with subarachnoid aneurysmal hemorrhage (SAH) receiving surgical clipping or endovascular coiling. Patients and methods: From Taiwan’s National Health Insurance Research Database, we enrolled patients with aneurysmal SAH who received clipping or coiling. After propensity score matching and adjustment for confounders, a generalized linear mixed model was used to determine significant differences in the accumulative hospital stay (days), intensive care unit (ICU) stay, and total medical cost for aneurysmal SAH, as well as possible subsequent surgical complications and recurrence. Results: The matching process yielded a final cohort of 8102 patients (4051 and 4051 in endovascular coil embolization and surgical clipping, respectively) who were eligible for further analysis. The mean accumulative hospital stay significantly differed between coiling (31.2 days) and clipping (46.8 days; *p* < 0.0001). After the generalized linear model adjustment of gamma distribution with a log link, compared with the surgical clipping procedure, the adjusted odds ratios (aOR; 95% confidence interval [CI]) of the medical cost of accumulative hospital stay for the endovascular coil embolization procedure was 0.63 (0.60, 0.66; *p* < 0·0001). The mean accumulative ICU stay significantly differed between the coiling and clipping groups (9.4 vs. 14.9 days; *p* < 0.0001). The aORs (95% CI) of the medical cost of accumulative ICU stay in the endovascular coil embolization group was 0.61 (0.58, 0.64; *p* < 0.0001). The aOR (95% CI) of the total medical cost of index hospitalization in the endovascular coil embolization group was 0·85 (0.82, 0.87; *p* < 0.0001). Conclusions: Medical resource consumption in the coiling group was lower than that in the clipping group.

## 1. Introduction

The prevalence of intracranial saccular aneurysms, as determined through radiographic and autopsy series, is estimated to be 3.2% in individuals without comorbidities, with a mean age of 50 years, and with a 1:1 gender ratio [1,2,3]. Among patients with cerebral aneurysms, 20–30% have multiple aneurysms [4]. Aneurysmal subarachnoid hemorrhage (SAH) occurs in the population at an estimated rate of 6–16 per 100,000 [5]. In North America, this rate translates into approximately 30,000 cases per year [6]. Thus, most aneurysms, particularly small aneurysms, do not rupture [6,7]. Rupture of an intracranial aneurysm is believed to account for 0.4–0.6% of all deaths [8]. Approximately 10% of patients die before reaching the hospital, and only one-third of patients have a satisfactory outcome after treatment. SAH is often a devastating event [9].

The most crucial goal of SAH management is the prevention of rebleeding by early repair of the unsecured aneurysm through surgical clipping or endovascular coiling. Surgical and endovascular techniques are available for treating aneurysms. A team of experienced surgeons and endovascular practitioners is responsible for selecting the therapy for treating a ruptured intracranial aneurysm. A subset of patients with aneurysmal SAH categorized as Hunt and Hess grades I–III may have more favorable outcomes after endovascular coiling [10,11]. The International Subarachnoid Aneurysm Trial (ISAT), the largest randomized trial concerning this medical condition, included 2143 patients with ruptured intracranial aneurysms and randomly assigned them to undergo neurosurgical clipping or endovascular coiling [12,13]. The findings revealed that short-term outcomes appeared to be more satisfactory after endovascular coiling compared with surgical clipping [10,11,12,13]. Many studies and one randomized trial (ISAT trial) have shown better survival outcomes in the patients with aneurysmal SAH receiving coiling [10,11,12,13]. Therefore, the issue of the patients’ wellbeing using coiling or clipping has been addressed. However, there is no answer for the financial cost of the two techniques for aneurysmal SAH. Thus, we conducted this study for long-term medical resource consumption between surgical clipping and endovascular coiling for aneurysmal SAH. The financial cost is very important for National health insurance policy formulation. The study results can provide policymakers with sufficient information to reconsider the whole reimbursement scheme from a holistic perspective.

In addition to unclear long-term clinical outcomes, long-term medical resource consumption, including accumulative hospital stay, accumulative intensive care unit (ICU) stay, and accumulative total medical cost of medicine use, recurrent aneurysm formation after endovascular coiling and surgical clipping in patients with aneurysmal SAH remains unclear. In the study, we estimated comprehensive medical resource consumption in patients with aneurysmal SAH who underwent endovascular coiling or surgical clipping.

## 2. Patients and Methods

### 2.1. Data Source

We conducted a population-based cohort study by using data from Taiwan’s National Health Insurance (NHI) Research Database (NHIRD). The NHIRD consists of all medical claims data regarding the disease diagnoses, procedures, drug prescriptions, demographics, and enrollment profiles of all beneficiaries of the NHI program [14]. From the NHIRD, we selected patients who had received a diagnosis of a ruptured intracranial aneurysm between 1 January 2011 and 31 December 2017. The follow-up period was from the index date to 31 December 2018. The index date was the therapeutic date of surgical clipping or endovascular coil embolization in the ruptured intracranial aneurysm cohort. Our protocols were reviewed and approved by the Institutional Review Board of Tzu-Chi Medical Foundation (IRB109-015-B). The Collaboration Center of Health Information Application of the NHIRD contains detailed treatment-related information regarding surgical procedures, in-hospital deaths [15,16], and whether surgical clipping or endovascular coil embolization was performed. The technique of endovascular coil embolization arm was all coiling, instead of stenting of the usage of a flow-diverter. Long-term medical resource consumption, including accumulative hospital stay, accumulative ICU stay, accumulative total medical costs of medicine use (e.g., long-term usage of antitrombotic medication), recurrent aneurysm formation after endovascular coiling, and surgical clipping were found in patients with aneurysmal SAH.

### 2.2. Study Cohort

Diagnoses of selected patients were confirmed on the basis of radiological data and whether they underwent surgical clipping or endovascular coil embolization. Inclusion criteria were the diagnosis of a new (first time) ruptured intracranial aneurysm, early aneurysm repair (within 24–72 h) in patients with low-grade SAH (Hunt and Hess grades I–III), and age ≥ 20 years. The key exclusion criterion was having had a ruptured intracranial aneurysm before the index date. In addition, we excluded patients with ruptured intracranial aneurysms who did not receive surgical clipping or endovascular coil embolization after diagnosis and those who received therapy more than 72 h after diagnosis [17]. The index hospitalization was defined as accumulative hospital days or accumulative ICU days of the first coiling or clipping and possibly the days following in the event of surgical complications and repetitive coiling or clipping for the recurrence of aneurysmal SAH. Finally, we categorized patients with ruptured intracranial aneurysms into the following groups for outcome comparison: group 1 (patients receiving endovascular coil embolization) and group 2 (patients receiving surgical clipping).

### 2.3. Covariates

Comorbidities were scored using the Charlson comorbidity index (CCI) [18,19]. Diabetes, congestive heart failure, hypertension, renal diseases, stroke, and transient ischemic attack (TIA) were not included in CCI scores to prevent duplicate weighting calculations when examining survival effects. Only comorbidities observed 6 months before the index date were included. Comorbid conditions were identified and included according to the diagnostic codes of the International Classification of Diseases, Ninth Revision, Clinical Modification (ICD-9-CM) for the first admission or the main diagnostic codes repeated more than twice for visits to the outpatient department. To reduce the effects of potential confounding factors when comparing treatment outcomes between groups, propensity score matching (PSM) was performed using a multivariate logistic regression model with the treatment group as the dependent variable and potential confounders as covariates. The PS was estimated using a multivariable logistic regression model, with the treatment groups and potential confounders representing dependent variables and covariates, respectively. The following confounders were included in the propensity score matching with the Mahalanobis metric (PSM-MM): age, sex, diagnosis year, location of aneurysm, diabetes, congestive heart failure, hypertension, renal diseases, stroke or TIA, CCI score, hospital level, hospital area, and income. All patients in the endovascular coil embolization group were matched at a 1:1 ratio with patients in the surgical clipping group through PSM and global optimization [20].

### 2.4. Endpoints

Dependent variables were the (1) accumulative hospital stay of index hospitalization and medical cost, (2) accumulative ICU stay of index hospitalization and medical cost, and (3) total medical cost of index hospitalization.

### 2.5. Statistical Analysis

This population-based retrospective cohort study was conducted using a generalized linear mixed model with multivariate analysis after adjustment for the covariates of age, sex, diagnosis year, aneurysm locations, diabetes, congestive heart failure, hypertension, renal diseases, stroke or TIA, CCI score, hospital level, hospital area, and income for all patients who underwent endovascular coil embolization or surgical clipping. Patients’ characteristics were first described according to the surgical modality by using descriptive statistics such as the mean and standard deviation for normal continuous data, medians and interquartile ranges for nonnormal continuous data, and number and proportions for categorical data. Student’s *t* test, analysis of variance, and their nonparametric counterpart tests were used, as appropriate. The multivariate mixed models accounting for patient clusters in hospitals were fitted to examine the effect of the therapeutic modality on outcomes. The generalized linear model of gamma distribution with a log link for the treatment of index hospitalization and medical cost was used after adjustment for the covariates. The significance level was set at 5%.

## 3. Results

### 3.1. Clinicopathological Characteristics

The initial included numbers of patients with low-grade SAH (Hunt and Hess grades I–III), and age ≥ 20 years were 11,047 patients in our study. The key exclusion criterion was having experienced a ruptured intracranial aneurysm before the index date (*N* = 94). In addition, we excluded patients with ruptured intracranial aneurysms who did not receive surgical clipping or endovascular coil embolization after diagnosis and those who received therapy more than 72 h after diagnosis (*N* = 63) [17]. There were 10,890 patients were enrolled in our population. Demographic and clinical parameters of patients with ruptured intracranial aneurysms before PSM are shown in Appendix A. All patients in the endovascular coil embolization group were matched at a 1:1 ratio with patients in the surgical clipping group through PSM and global optimization [20]. The matching process yielded a final cohort of 8102 patients (4051 and 4051 in endovascular coil embolization and surgical clipping, respectively) who were eligible for further analysis; their characteristics are summarized in Table 1. After PSM, the covariates of age, sex, diagnosis year, aneurysm location, diabetes, congestive heart failure, hypertension, renal diseases, stroke or TIA, CCI score, hospital level, hospital area, and income were observed to be similar in the two cohorts (Table 1). All standardized differences for each covariate listed in Table 1 were <0.1, indicating a balanced distribution between the two treatment groups. No significant difference in in-hospital deaths was noted between the two treatments. The use of a well-matched PSM design can reduce the selection bias in different treatments for patients with ruptured intracranial aneurysms (see Table 1). The mean follow-up time were 53.7 and 51.3 months for endovascular coil embolization and surgical clipping, respectively.

### 3.2. Accumulative Hospital Stay of Index Hospitalization and Medical Cost Stratified by Coiling or Clipping

Table 2 lists the accumulative hospital stay (days) of index hospitalization stratified by surgical modality. The mean accumulative hospital stay significantly differed between the coiling (31.2 days) and clipping (46.8 days) groups (*p* < 0.0001). After the generalized linear model adjustment of gamma distribution with a log link, compared with the surgical clipping group, the adjusted odds ratio (aOR; 95% confidence intervals [CI]) of the medical cost of accumulative hospital stay in the endovascular coil embolization group was 0.63 (0.60, 0.66; *p* < 0.0001).

### 3.3. Accumulative Intensive Care Unit Stay of Index Hospitalization and Medical Cost Stratified by Coiling or Clipping

Table 3 presents the accumulative ICU stay of index hospitalization stratified by surgical modality. The mean accumulative ICU stay significantly differed between the coiling and clipping groups (9.4 vs. 14.9 days; *p* < 0.0001). After the generalized linear model adjustment of gamma distribution with a log link, compared with the surgical clipping group, the aOR (95% CI) of the medical cost of accumulative ICU stay in the endovascular coil embolization group was 0.61 (0.58, 0.64; *p* < 0.0001).

### 3.4. Total Medical Cost of Index Hospitalization Stratified by Coiling or Clipping

Table 4 shows the total medical cost of index hospitalization stratified by surgical modality. The mean total medical cost was NTD 608,863.7 and NTD 517,299.0 for the coiling and clipping groups, respectively; the cost significantly differed between the two groups (*p* < 0.0001). After the generalized linear model adjustment of gamma distribution with a log link, compared with the surgical clipping group, the aOR (95% CI) of the total medical cost of index hospitalization in the endovascular coil embolization group was 0.85 (0.82, 0.87; *p* < 0.0001).

## 4. Discussion

The long-term outcomes of medical cost remain unclear, although some studies have reported that the equivocal short-term cost might be lower or equal between coiling and clipping in aneurysmal SAH [21,22,23,24]. No long-term study has estimated the accumulative hospital stay, accumulative ICU stay, or accumulative total medical cost between coiling and clipping in patients with a new aneurysmal SAH, as well as their possible subsequent surgical complications and recurrence. It is true that the higher medical resources consumption does not necessarily mean a higher recurrence rate. However, our endpoint was long-term medical resource consumption, instead of the recurrence rate. We focused on the endpoint of accumulative financial cost of coiling and clipping for aneurysmal SAH. The accumulative finical cost of recurrence of re-clipping or re-coiling for recurrence was calculated in our study. The incidence of recurrence of aneurysmal SAH between coiling and clipping has been reported by many studies and there was an increased risk of recurrent bleeding from a coiled aneurysm compared with a clipped aneurysm, but the risks were small [8,25,26]. Thus, we calculated the total medical cost for aneurysmal SAH and possible subsequent surgical complications and recurrence, instead of the recurrence rate, since there was no novelty for the incidence of recurrence of aneurysmal SAH between coiling and clipping. In our current study, we examined the total comprehensive medical resource consumption between coiling and clipping for low-grade aneurysmal SAH. Our findings can serve as the reference for implementing future health policies if no selection bias exists for treatment or inferior survival outcomes.

Until now, no study has examined long-term total comprehensive medical resource consumption between coiling and clipping for low-grade aneurysmal SAH. Some studies have compared hospital stay, ICU stay, or medical cost between coiling and clipping for unruptured intracranial aneurysms [27,28,29,30,31,32]. In patients with unruptured intracranial aneurysms, findings regarding total index hospitalization costs for clipping or coiling have been controversial [27,28,29,30,31,32]. No solid conclusions regarding the medical cost of coiling and clipping in patients with unruptured intracranial aneurysms are available [27,28,29,30,31,32]. In patients with ruptured intracranial aneurysms, a short-term ISAT study with subgroup analysis demonstrated no significant differences in costs between the endovascular and neurosurgery groups at either the 12- or 24-month follow-up [33]. In addition, other short-term retrospective studies have reported controversial data regarding medical costs in the treatment of coiling or clipping for patients with ruptured intracranial aneurysms [21,22,23,24]. However, retrospective studies with a short-term follow-up and small sample size have indicated that the length of stay for endovascular coiling was much shorter than that for neurosurgical clipping and decreased over time [21,24]. Nevertheless, no study comparing comprehensive medical resource consumption with a sufficient follow-up duration and sample size has examined the total medical cost for patients with ruptured intracranial aneurysms. Our data showed that long-term hospital stay and ICU stay were shorter in the coiling group than in the clipping group (Table 2 and Table 3); these findings are compatible with those of previous short-term studies [21,24,31]. The total medical cost was lower in the coiling group than in the clipping group (Table 4); this finding can be attributed to a shorter hospital stay and ICU stay in the coiling group (Table 1 and Table 3).

The financial cost should not be the neurosurgeon’s main concern, the bias is the initial decision for coiling or clipping by the neurosurgeon. However, there is no strong association of selection bias of coiling or clipping for aneurysmal SAH, except age, aneurysm size, greater experience (academic centers or non-academic centers), or location of the aneurysm [12,17,25,34,35], which were adjusted or controlled in the enrolled patients. Diagnoses of selected patients were confirmed on the basis of radiological data and whether they underwent surgical clipping or endovascular coil embolization. Inclusion criteria were the diagnosis of a new (first time) ruptured intracranial aneurysm, early aneurysm repair (within 24–72 h) in patients with low-grade SAH (Hunt and Hess grades I–III), and age ≥ 20 years. As shown in Table 1, no imbalance in covariates was observed between the coiling and clipping groups for aneurysmal SAH. The balance in covariates between the two treatments might not have resulted in selection bias for treatment in our patients with aneurysmal SAH (Table 1). In our findings (Table 1 or Appendix A), female gender is the predominant factor for the occurrence of aneurysmal subarachnoid hemorrhage in Taiwan, which is compatible with the previous studies [36,37]. The gender-related differences in aneurysmal subarachnoid hemorrhage seem similar to both Western and Eastern countries [36,37]. We also observed significantly superior survival rates in the coiling group compared with the clipping group. The superior survival rates observed in the coiling group are compatible with the findings of previous studies [12,13]. Although all-cause deaths in the coiling group were fewer than those in the clipping group, the total medical cost was lower in the coiling group (Table 4). Because medical cost is only measured for living patients, longer survival increases medical costs. However, the total medical cost was still lower in the coiling group. Moreover, because survival was longer and the total medical cost was lower in the coiling group, coiling was more cost-effective compared with clipping for patients with ruptured intracranial aneurysms. Our head-to-head PSM design study including a long-term follow-up is the first to report that coiling treatment is more cost effective compared with clipping treatment in aneurysmal SAH.

As shown in Table 2, the long-term cumulative hospital stay for aneurysmal SAH and possible subsequent surgical complications and recurrence was shorter in the coiling treatment group than in the clipping treatment group (Table 2). Owing to a shorter hospital stay in the coiling treatment group, the corresponding median cost of hospitalization was lower in the coiling group. Our findings are compatible with those of previous short-term studies [22,24,31]. As shown in Table 3, the long-term ICU stay for aneurysmal SAH and possible subsequent surgical complications and recurrence was shorter in the coiling treatment group than in the clipping treatment group, with the corresponding medical cost of ICU care being lower in the coiling treatment group. Our findings are compatible with those of a short-term, small-sample observational study that determined that compared with neurosurgical clipping, coiling was associated with significant benefits in terms of a decreased need for ICU care in patients with ruptured intracranial aneurysms [21]. This is the first head-to-head PSM design study including a long-term follow-up and the largest sample size to report shorter accumulative hospital stay, ICU stay, and lower total medical cost in the coiling group compared with the clipping group. The cost effectiveness of coiling treatment was superior to clipping treatment in patients with low-grade aneurysmal SAH (Table 1, Table 2, Table 3 and Table 4).

The strength of our study is that covariates were balanced between coiling and clipping groups. We believe that the selection bias of coiling or clipping was controlled (Table 1). Furthermore, our study included the largest sample size and the longest long-term follow-up for evaluating medical resource consumption in terms of accumulative hospital stay, ICU stay, and total medical cost for treatment of aneurysmal SAH and possible subsequent surgical complications and recurrence between coiling and clipping. Our study findings revealed a shorter cumulative hospital stay and ICU stay in the coiling group for aneurysmal SAH and possible subsequent surgical complications and recurrence (Table 3). Moreover, total medical cost for aneurysmal SAH and possible subsequent surgical complications and recurrence was lower in the coiling group (Table 4). The total benefits of lower medical resource consumption in patients with aneurysmal SAH receiving coiling compared with those receiving clipping can be used for establishing further health policies. The cost effectiveness was superior in the coiling group with a longer survival rate after PSM (Table 1). The study results can provide policymakers with sufficient information to reconsider the whole reimbursement scheme from a holistic perspective.

Our study has some limitations. First, it was a retrospective observational study wherein patients with aneurysmal SAH were nonrandomly assigned to each treatment type based on the preferences of clinicians. Second, although no significant differences in covariate distribution was observed between coiling and clipping (Table 1), unobserved confounders may have led to biased results. Third, despite the lower total use of medical resources in the coiling group through PSM and covariates controlled, in some cases the choice of the surgical treatment method is dictated by the unfavorable location and morphology of the aneurysm (large aneurysm size, pathologic shape) and intracerebral hematoma, which means that the surgical clipping gives better results, and then the financial considerations are not so important. Finally, cost data can vary substantially between countries. Hence, our study findings may not be generalized. Despite these limitations, our population-based nationwide analysis provides updated, long-term, follow-up information regarding medical resource consumption for coiling compared with clipping treatment. Information concerning medical resource consumption for various treatments is crucial for establishing future health policies for patients with aneurysmal SAH.

## 5. Conclusions

After PSM and adjustment for confounders, medical resource consumption in the coiling group was lower than that in the clipping group.

## Figures and Tables

**Table 1 ijerph-18-05989-t001:** Demographic and clinical parameters of propensity score–matched patients with ruptured intracranial aneurysms.

		**Endovascular Coil Embolization** ***N* = 4051**		**Surgical Clipping** ***N* = 4051**		**Standardized Difference**
		***n***	**(%)**	***n***	**(%)**	
Age (years)	Mean (SD)	58.0	(13.7)	57.5	(13.1)	0.035
	Median (Q1–Q3)	58	(49–68)	57	(49–67)	
	20–64	2716	(67.0)	2839	(70.1)	0.028
	65–74	791	(19.5)	746	(18.4)	0.047
	75–84	474	(11.7)	415	(10.2)	0.039
	85+	70	(1.7)	51	(1.3)	0.035
Gender	Male	1479	(36.5)	1432	(35.3)	0.024
	Female	2572	(63.5)	2619	(64.7)	
Treatment year	2011–2013	1412	(34.9)	1481	(36.6)	0.012
	2014–2015	1172	(28.9)	1194	(29.5)	0.047
	2016–2017	1467	(36.2)	1376	(34.0)	0.042
Location of aneurysm	ACA	1860	(45.9)	1860	(45.9)	0.000
	ICA	376	(9.3)	376	(9.3)	0.000
	MCA	1372	(33.9)	1372	(33.9)	0.000
	VBA	302	(7.5)	302	(7.5)	0.000
	PCA	141	(3.5)	141	(3.5)	0.000
Diabetes	No	3437	(84.8)	3497	(86.3)	0.040
	Yes	614	(15.2)	554	(13.7)	
Congestive heart failure	No	3945	(97.4)	3962	(97.8)	0.027
	Yes	106	(2.6)	89	(2.2)	
Hypertension	No	1895	(46.8)	1983	(49.0)	0.043
	Yes	2156	(53.2)	2068	(51.0)	
Renal diseases	End-stage renal disease	45	(1.1)	32	(0.8)	0.038
	Chronic kidney disease	138	(3.4)	118	(2.9)	0.028
	No renal diseases	3868	(95.5)	3901	(96.3)	0.041
Stroke or TIA	No	2440	(60.2)	2616	(64.6)	0.090
	Yes	1611	(39.8)	1435	(35.4)	
CCI Score	0	628	(15.5)	683	(16.9)	0.013
	1	2304	(56.9)	2353	(58.1)	0.024
	2+	1119	(27.6)	1015	(25.1)	0.058
Hospital level	Academic centers	3178	(78.4)	3198	(78.9)	0.012
	Nonacademic centers	873	(21.6)	853	(21.1)	
Hospital area	North	2329	(57.5)	2172	(53.6)	0.055
	Center	735	(18.1)	843	(20.8)	0.067
	South	871	(21.5)	915	(22.6)	0.026
	East	116	(2.9)	121	(3.0)	0.007
Income level	<NTD 18,000	924	(22.8)	872	(21.5)	0.026
	NTD 18,000–22,500	842	(20.8)	903	(22.3)	0.037
	NTD 22,500–30,000	915	(22.6)	908	(22.4)	0.004
	NTD 30,000+	1370	(33.8)	1368	(33.8)	0.001
		**Endovascular Coil Embolization** ***N* = 4051**		**Surgical Clipping** ***N* = 4051**		***p* Value**
		***N***	**(%)**	***N***	**(%)**	
All-cause death		736	(18.2)	812	(20.0)	0.0294
Follow-up time, months	Mean (SD)	53.7	(23.5)	51.3	(24.2)	<0.0001
	Median (Q1–Q3)	53.0	(30–74)	49.7	(31–71)	

CCI, Charlson comorbidity index; SD, standard deviation; IQR, interquartile range; n, number; NTD, New Taiwan dollar; ICA, internal carotid artery; ACA, anterior cerebral artery; MCA, middle cerebral artery; PCA; posterior cerebral artery; VBA, vertebral basilar artery; TIA, transient ischemic attack.

**Table 2 ijerph-18-05989-t002:** Accumulative hospital stay during index hospitalization and medical cost stratified by aneurysm repair modalities in propensity score–matched patients with ruptured cerebral aneurysms who received surgical clipping or endovascular coiling.

Hospital Stay during Index Hospitalization, Days					Generalized Linear Model of Gamma Distribution with a Log Link for Medical Cost *				
Treatment Modality	Mean	(SD)	*p* Value	Median	B	S.E.	Exp(b)	(95% CI)	*p* Value
<0.0001									<0.0001
Surgical clipping (ref.)	46.8	(109.5)		23	ref.				
Endovascular coiling	31.2	(85.1)		14	−0.4607	0.0232	0.63	(0.60, 0.66)	

* Models with covariates of age, sex, treatment year, diabetes, congestive heart failure, hypertension, renal diseases, previous stroke or TIA symptoms, CCI score, hospital level, hospital area, and income. SD, standard deviation; CI, confidence interval; ref.; reference group b: This is the coefficient for the constant (also called the “intercept”) in the null model. S.E.: This is the standard error around the coefficient for the constant. Exp(b): This is the exponentiation of the “b” coefficient, which is an odds ratio.

**Table 3 ijerph-18-05989-t003:** Accumulative intensive care unit stay during index hospitalization and medical cost stratified by aneurysm repair modalities in propensity score–matched patients with ruptured cerebral aneurysms who received surgical clipping or endovascular coiling.

ICU Stay during Index Hospitalization, Days					Generalized Linear Model of Gamma Distribution with a Log Link for Medical Cost *				
Treatment Modality	Mean	(SD)	*p* Value	Median	b	S.E.	Exp(b)	(95% CI)	*p* Value
			<0.0001						<0.0001
Surgical clipping (ref.)	14.9	(16.7)		10	ref.				
Endovascular coiling	9.4	(13.3)		4	−0.4932	0.0230	0.61	(0.58, 0.64)	

* Models with covariates of age, sex, treatment year, diabetes, congestive heart failure, hypertension, renal diseases, previous stroke or TIA symptoms, CCI score, hospital level, hospital area, and income. SD, standard deviation; CI, confidence interval; ref.; reference group b: This is the coefficient for the constant (also called the “intercept”) in the null model. S.E.: This is the standard error around the coefficient for the constant. Exp(b): This is the exponentiation of the “b” coefficient, which is an odds ratio.

**Table 4 ijerph-18-05989-t004:** Total medical cost of index hospitalization stratified by aneurysm repair modalities in propensity score–matched patients with ruptured cerebral aneurysms who received surgical clipping or endovascular coiling.

Total Medical Cost (NTD)					Generalized Linear Model of Gamma Distribution with a Log Link for Medical Cost *				
Treatment Modality	Mean	(SD)	*p* Value	Median	b	S.E.	Exp(b)	(95% CI)	*p* Value
			<0.0001						<0.0001
Surgical clipping (ref.)	608,863.7	(611,619.9)		412,536	ref				
Endovascular coiling	517,299.0	(522,550.5)		372,335	−0.1660	0.0158	0.85	(0.82, 0.87)	

* Models with covariates of age, sex, treatment year, diabetes, congestive heart failure, hypertension, renal diseases, previous stroke or TIA symptoms, CCI score, hospital level, hospital area, and income. SD, standard deviation; NTD, New Taiwan dollar; CI, confidence interval; ref.; reference group b: This is the coefficient for the constant (also called the “intercept”) in the null model. S.E.: This is the standard error around the coefficient for the constant. Exp(b): This is the exponentiation of the “b” coefficient, which is an odds ratio.

## Data Availability

The data sets supporting the study conclusions are included in this manuscript and its Appendix A.

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
