# Peer review of "Long-Term Medical Resource Consumption between Surgical Clipping and Endovascular Coiling for Aneurysmal Subarachnoid Hemorrhage: A Propensity Score–Matched, Nationwide, Population-Based Cohort Study"

_ijerph, 2021, doi:10.3390/ijerph18115989_

Round 1

Reviewer 1 Report

Comments to the Authors

In manuscript "Long-Term Medical Resource Consumption Between Surgical Clipping and Endovascular Coiling for Aneurysmal Subarachnoid Hemorrhage: Propensity Score–Matched, Nationwide,  Population-Based Cohort Study", the authors estimate long-term medical resource consumption in patients with subarachnoid aneurysmal hemorrhage (SAH) receiving surgical clipping or endovascular coiling.

The population-based retrospective cohort study is interesting and important. Based on their research, the authors conclude that the total medical resource consumption in the coiling group was lower in terms of accumulative hospital stay, ICU stay, and total medical cost than in the clipping group.

The manuscript requires minor corrections:

  1. In the Introduction section, there is no clear indication of the purpose of the research undertaken. Please expand and complete.
  2. Table 1 contains a lot of data, please to reformat it to make it easier to read, some of the data, for example "n" in the "Surgical clipping" group are illegible.
  3. The table numbers described in the text do not agree with the actual numbers of the tables presented.

Table 3 shows the total medical cost of index hospitalization stratified by surgical (...) Should be: Table 4 shows the total medical cost of index hospitalization stratified by surgical (...).

  1. In the discussion part it should be mentioned that despite the lower total use of medical resources in the coiling group, in some cases the choice of the surgical treatment method is dictated by the unfavorable location and morphology of the aneurysm (large aneurysm size, pathologic shape) and intracerebral hematoma, which means that the surgical clipping gives better results, and then the financial considerations are not so important.

Author Response

Reviewer 1

In manuscript "Long-Term Medical Resource Consumption Between Surgical Clipping and Endovascular Coiling for Aneurysmal Subarachnoid Hemorrhage: Propensity Score–Matched, Nationwide,  Population-Based Cohort Study", the authors estimate long-term medical resource consumption in patients with subarachnoid aneurysmal hemorrhage (SAH) receiving surgical clipping or endovascular coiling.

The population-based retrospective cohort study is interesting and important. Based on their research, the authors conclude that the total medical resource consumption in the coiling group was lower in terms of accumulative hospital stay, ICU stay, and total medical cost than in the clipping group.

The manuscript requires minor corrections:

Response: Thank you for your appreciation and improve our manuscript a lot.

In the Introduction section, there is no clear indication of the purpose of the research undertaken. Please expand and complete.

Response: Thank you very much

According to your suggestions, we have rewritten the statements of clear indication of the purpose of the research undertaken in the section of discussion.

There were many studies and one randomized trial (ISAT trial) have shown better survival outcomes in the patients with aneurysmal SAH receiving coiling.[3,7-9] Therefore, the issue of the patients’ wellbeing using coiling or clipping have been addressed. However, there is no answer for the financial cost of the two techniques for aneurysmal SAH. Thus, we conducted the study for long-term medical resource consumption between surgical clipping and endovascular coiling for aneurysmal SAH. The financial cost is very important for National health insurance policy formulation. The study results can provide policymakers with sufficient information to reconsider the whole reimbursement scheme from a holistic perspective.

Table 1 contains a lot of data, please to reformat it to make it easier to read, some of the data, for example "n" in the "Surgical clipping" group are illegible.

Response: Thank you

We have tried our best to make Table 1 easier to read. We have tried our best to alignment the numbers and words in Table 1. We also added IQR in the Table 1.

The table numbers described in the text do not agree with the actual numbers of the tables presented.

Table 3 shows the total medical cost of index hospitalization stratified by surgical (...) Should be: Table 4 shows the total medical cost of index hospitalization stratified by surgical (...).

Response: Thank you very much

     We have checked the table numbers described in the text matched with the actual numbers of the tables presented throughout the manuscript.

In the discussion part it should be mentioned that despite the lower total use of medical resources in the coiling group, in some cases the choice of the surgical treatment method is dictated by the unfavorable location and morphology of the aneurysm (large aneurysm size, pathologic shape) and intracerebral hematoma, which means that the surgical clipping gives better results, and then the financial considerations are not so important.

Response: Thank you for your valuable suggestions very much

According to your suggestions, we have added the statements in the section of discussion as followings:

      Despite the lower total use of medical resources in the coiling group through PSM and covariates controlled, in some cases the choice of the surgical treatment method is dictated by the unfavorable location and morphology of the aneurysm (large aneurysm size, pathologic shape) and intracerebral hematoma, which means that the surgical clipping gives better results, and then the financial considerations are not so important.

Reviewer 2 Report

The data, presented in the paper, is about the cost-effectiveness of initial hospitalisation for treatment of aneurysmal SAH. Microsurgical clipping and endovascular coiling were compared. The name of the article and data analysis are misleading to conclusion, that a long-term follow-up is included in the study. As I can understand from the text, this was a long period of inclusion (2011-2017), not a long term of follow-up - no data about the re-hospitalisation, re-bleeding, long term medication usage is presented. There are multiple publications, that claim the superiority of long-term durability of clip vs. coil, that gives the series of patients with re-bleeding, re-coiling, what is connected with the increase in expenses. Furthermore, what methods of endovascular treatment was used - coiling, stenting of usage of flow-diverters. The expenses of technology used may alter the final calculations, and usage of these devices is connected with long-term usage of antitrombotic medication.

If the follow-up was used, the follow-up was finished in one year (December 31, 2018) after inclusion was closed. And about 35% of the patients were included on year 2016-2017. One to two years of follow-up is definitely not sufficient period for "long-term follow-up" and cost-effectiveness.

The extensive use of references (15-23) not directly connected to the subject must be reduced.

Line 31 - enrolled MALE patients. As I can understand males and females were included

My suggestion is to revise the interpretation of long-term follow-up. Or to add the follow-up data and it's analysis.

Author Response

Reviewer 2

The data, presented in the paper, is about the cost-effectiveness of initial hospitalisation for treatment of aneurysmal SAH. Microsurgical clipping and endovascular coiling were compared. The name of the article and data analysis are misleading to conclusion, that a long-term follow-up is included in the study. As I can understand from the text, this was a long period of inclusion (2011-2017), not a long term of follow-up - no data about the re-hospitalisation, re-bleeding, long term medication usage is presented. There are multiple publications, that claim the superiority of long-term durability of clip vs. coil, that gives the series of patients with re-bleeding, re-coiling, what is connected with the increase in expenses. Furthermore, what methods of endovascular treatment was used - coiling, stenting of usage of flow-diverters. The expenses of technology used may alter the final calculations, and usage of these devices is connected with long-term usage of antitrombotic medication.

Response: Thank you very much

      The technique of coiling arm was all coiling, instead of stenting of usage of flow-diverter. All medical cost also include medication use like long-term usage of antitrombotic medication. We also written the sentences in the section of method. 

      According to your suggestions, we added the follow-up time in Table 1 to prove the long-term follow-up in our study. The mean follow-up time were 53.7 and 51.3 months for endovascular coil embolization and surgical clipping, respectively. If the reviewer and Editor both think correction of the “long-term follow-up” as “long period of inclusion”, we will correct the associated sentences throughout the manuscript.

       It is true that the higher medical resources consumption doesn’t necessary mean higher recurrence rate. However, our endpoint is long-term medical resource consumption, instead of recurrence rate. We focus on the endpoint of accumulative financial cost of coiling and clipping for aneurysmal SAH. The accumulative finical cost of recurrence of re-clipping or re-coiling for recurrence had been calculated in our study. The incidence of recurrence of aneurysmal SAH between coiling and clipping have been reported by many studies and there was an increased risk of recurrent bleeding from a coiled aneurysm compared with a clipped aneurysm, but the risks were small [5,10,11]. Thus, we calculated total medical cost for aneurysmal SAH and possible subsequent surgical complications and recurrence, instead of recurrence rate; because there was no novelty for the incidence of recurrence of aneurysmal SAH between coiling and clipping.

The index hospitalization was defined as accumulative hospital days or accumulative ICU days of the first coiling or clipping and possibly the days following in the event of surgical complications and repetitive coiling or clipping for the recurrence of aneurysmal SAH. No long-term study has estimated the accumulative hospital stay, accumulative ICU stay, or accumulative total medical cost between coiling and clipping in patients with a new aneurysmal SAH as well as their possible subsequent surgical complications and recurrence. In our current study, we examined the total comprehensive medical resource consumption between coiling and clipping for low-grade aneurysmal SAH. Our findings can serve as the reference for implementing future health policies if no selection bias exists for treatment or inferior survival outcomes.   

If the follow-up was used, the follow-up was finished in one year (December 31, 2018) after inclusion was closed. And about 35% of the patients were included on year 2016-2017. One to two years of follow-up is definitely not sufficient period for "long-term follow-up" and cost-effectiveness.

Response: Thank you very much

According to your suggestions, we added the follow-up time in Table 1 to prove the long-term follow-up in our study. If the reviewer and Editor both think “long-term” is ot suitable, we will delete the “long-term”. The mean follow-up time were 53.7 and 51.3 months for endovascular coil embolization and surgical clipping, respectively.

The extensive use of references (15-23) not directly connected to the subject must be reduced.

Response: Thank you very much

      We have reduced the references (15-23).

Line 31 - enrolled MALE patients. As I can understand males and females were included

Response: thank you

   We have revised the typo as followings:

From Taiwan’s National Health Insurance Research Database, we enrolled patients with aneurysmal SAH who received clipping or coiling.

My suggestion is to revise the interpretation of long-term follow-up. Or to add the follow-up data and it's analysis.

Response: Thank you very much

According to your suggestions, we added the follow-up time in Table 1 to prove the long-term follow-up in our study. The mean follow-up time were 53.7 and 51.3 months for endovascular coil embolization and surgical clipping, respectively.

Round 2

Reviewer 2 Report

The corrections were made, that make the paper's ideas clear and based on the data.